# Impact of the COVID-19 Pandemic on the Diagnosis and Management of Non-Melanoma Skin Cancer in the Head and Neck Region: A Retrospective Cohort Study

**DOI:** 10.3390/healthcare12040501

**Published:** 2024-02-19

**Authors:** Simone Benedetti, Andrea Frosolini, Lisa Catarzi, Agnese Marsiglio, Paolo Gennaro, Guido Gabriele

**Affiliations:** Maxillofacial Surgery Unit, Department of Medical Biotechnologies, University of Siena, 53100 Siena, Italy; simone.benedetti@student.unisi.it (S.B.); lisa.catarzi@student.unisi.it (L.C.); agnese.marsiglio@student.unisi.it (A.M.); paolo.gennaro@unisi.it (P.G.);

**Keywords:** COVID-19, skin cancer, squamous cell carcinoma, head and neck, dermatology, oncology

## Abstract

The present study investigates the impact of the COVID-19 pandemic on the management of Non-Melanoma Skin Cancer (NMSC) in the head and neck region. Conducted at the University Hospital “Le Scotte” in Siena, Italy, the research includes 111 patients treated from 2018 to 2021. The study aims to understand how pandemic-related healthcare changes affected NMSC treatment, focusing on differences in diagnosis and management before and during the pandemic. Methods involved retrospective analysis of patient demographics, clinical characteristics, lesion details, and treatment modalities, using Jamovi software (version 1.6) for statistical analysis. Results revealed the scalp as the most common NMSC site, with Squamous Cell Carcinoma (SCC) being the predominant histotype. A significant rise in Basal Cell Carcinoma (BCC) cases and a reduction in surgery duration were noted during the pandemic. The shift to local anesthesia was more pronounced, reflecting the necessity to adapt to healthcare limitations. Despite the disruptions caused by the pandemic, there was no significant drop in NMSC cases, which is attributed to the noticeable nature of head and neck lesions. In conclusion, this study highlights that the COVID-19 pandemic significantly influenced surgical practices in NMSC management, emphasizing the need for effective healthcare strategies that balance quality patient care with public health safety measures.

## 1. Introduction

Skin cancer, one of the most prevalent malignancies worldwide, comprises a heterogeneous group of neoplasms originating from the skin’s various cell types. Non-Melanoma Skin Cancer (NMSC), predominantly encompassing Basal Cell Carcinoma (BCC) and Squamous Cell Carcinoma (SCC), represents the most common form of skin cancer, primarily affecting individuals with a history of chronic sun exposure or other predisposing factors. These malignancies are particularly prevalent in regions with high levels of ultraviolet radiation, such as the Mediterranean area, where the picturesque landscape and warm climate often result in extended sun exposure [1]. The head and neck are the most commonly affected areas, with a prevalence of over 85%, necessitating multidisciplinary management, including the involvement of specialized maxillofacial surgeons for radical treatment, functional preservation, and aesthetic reconstruction [2,3].

The outbreak of the novel coronavirus disease 2019 (COVID-19), caused by the severe acute respiratory syndrome coronavirus 2 (SARS-CoV-2), has dramatically altered the landscape of healthcare worldwide. Since its emergence in early 2020, the pandemic has placed immense strain on healthcare systems, forcing prioritization and reallocation of resources towards the management of COVID-19 patients [4,5,6]. Despite the efforts to maintain pre-pandemic response times, a variety of medical specialties, including both acute and long-term care, have reported delays in diagnosis and treatment due to strained hospital systems, COVID-19 precautions, and workforce redistribution. For instance, while some authors reported evidence of surgical de-escalation for cancer treatment, in favor of nonsurgical therapy, especially during the initial phases of the pandemic [7], others noticed a sharp decline in cancer screening and treatment, suggesting a delayed effect of the pandemic on cancer management [8,9]. A recently published study analyzed the changes in the number and stage distribution of new cancer diagnoses between 2019 and 2020 in the USA. Based on the results on 405,050 adults, a significant underdiagnosis of cancer and a reduction in early-stage cancer diagnoses, especially among populations with limited medical access, was reported in 2020. According to the authors, there is a need to track the pandemic’s extended impact on cancer-related morbidity, survival rates, and mortality over time [10].

In regard to NMSC, previous studies have reported discordant results, with some authors reporting a significant impact of the pandemic on treatment delay [11,12], while other authors reported no significant changes [13]. In this challenging context of the pandemic, the relationship between surgeons and anesthesiologists becomes even more important than before, given the need to optimize resources such as time and workforce [14]. This dyad plays a crucial role in keeping the work in the surgical theatre on schedule, making important choices on patient management that range from the type of anesthesia (local, plexus, or general) to the invasiveness of the surgical procedure, which must be compatible with the patient’s conditions.

The primary aim of this retrospective series is to report our experience over the last four years in managing NMSC of the head and neck in a tertiary referral maxillofacial center. A secondary aim is to highlight any differences regarding diagnosis and management before and during the COVID-19 pandemic. In the realm of oncological surgery, the unprecedented challenges posed by the pandemic have necessitated a re-evaluation of surgical priorities and practices. By focusing on NMSC, a prevalent yet often underrepresented area in cancer research, this study provides insights into how surgical management has adapted in response to this healthcare challenge.

## 2. Materials and Methods

To address the research purposes, we performed a retrospective review of the clinical features and the surgical management of patients affected by Non-Melanoma Skin Cancer (NMSC) in the head and neck district, treated by the maxillofacial surgery unit of the University Hospital “Le Scotte” in Siena, Italy, between January 2018 and December 2021. The study protocol was designed in conformity with the ethical guidelines of the 1975 Declaration of Helsinki and was approved by the Ethical Committee for clinical research of the University Hospital of Siena (approval no. 16/2023). All data are reported according to the STROBE guidelines.

### 2.1. Study Structure

Patients treated for NMSC who underwent surgery between 1 January 2018, and 31 December 2021, were included in the study and divided into two cohorts: one comprising patients treated during the 2018–2019 period (named the pre-pandemic cohort), and the other including patients treated during the 2020–2021 period (named the pandemic cohort).

### 2.2. Inclusion Criteria

Patients with incomplete data or those who died before treatment were excluded from the study. Each patient underwent a preoperative biopsy to confirm the diagnosis of NMSC. The collected data for each patient included (i) personal data, (ii) patient’s history (presentation of the present skin lesion, past medical history, family history, occupational background, present and past use of medication), (iii) clinical characteristics of the lesion (dimensions, location, TNM classification), (iv) biopsy data, (v) reconstructive technique (primary intention closure, use of dermal regeneration templates, local flap), (vi) surgery duration, (vii) histological features (histotype, differentiation, depth of invasion—DOI, margin assessment), (viii) approach to recurrences, and (ix) clinical and radiological follow-up data.

### 2.3. Statistical Analysis

Data from both cohorts were then compared to assess any differences between the two groups. Quantitative variables were reported as mean ± standard deviation (SD) and median values. When appropriate, variables were dichotomized according to median values. The statistical analyses were performed using Jamovi software (version 1.6, 2021, open access software available at https://www.jamovi.org, accessed on 24 November 2022), as previously reported [5]. The chi-square test, the Mann–Whitney U test, the Spearman’s test, the Kaplan–Meier method, and the Cox proportional hazards model were used as appropriate. A *p*-value < 0.05 was considered significant.

## 3. Results

### 3.1. Epidemiological, Clinical, and Surgical Characteristics of Included Patients

Following inclusion and exclusion criteria, 111 patients were enrolled in the study (66 males and 45 females, mean age 79 ± 11.99 years). Considering multiple lesions in a single patient, which occurred in 24 cases (21.6%), the population accounted for 160 NMSC cases. The most frequent location was the scalp, involved in 46 cases (28.7%). The relative frequency of different locations is reported in Table 1. The mean dimension of the lesions was 2.36 ± 1.7 cm. According to the TNM clinical classification, there were 83 T1 (53.2%), 43 T2 (27.6%), and 21 T3 cases (19.2%). We observed three N positive cases (2%) and all included cases were M0. The definitive diagnosis was SCC in 87 cases (54.4%), BCC in 66 cases (41.2%), and other histotypes (Merkel cell carcinoma, atypical fibroxanthoma, and microcystic adnexal carcinoma) in 7 cases (4.4%). The mean depth of invasion (DOI) was 4.59 ± 4 mm. For SCC cases, the differentiation was well differentiated in 17 cases, (28.8%), moderately differentiated in 34 cases (57.6%), and poorly differentiated in 3 cases (5.1%). Local anesthesia was performed in 73 cases (45.6%), whereas general anesthesia was performed in 79 cases (49.4%). The type of anesthesia was not reported in 9 cases (5%). Tumor margins were negative in 120 cases (77.4%) and positive in 35 cases (22.6%).

### 3.2. Comparison between Pre-Pandemic and Pandemic Cohorts

Overall, we observed 63 cases in the pre-pandemic cohort and 97 cases in the pandemic cohort. Conducting a sub-analysis comparison (see Table 2 and Table 3), we identified a significant increase in BCC diagnosis, rising from 20 to 46 cases (*p* = 0.037) during the pandemic. Moreover, a statistically significant change in surgical time was retrieved, with a mean of 159.333 ± 100.097 min in the pre-pandemic period versus 125.415 ± 103.848 min in the pandemic period (*p* = 0.003). Surgical time was found to be positively correlated with cancer dimension in the 2020–2021 cohort, as shown in Figure 1 (0.315 Spearman’s rho, *p*-value = 0.002), but not in the 2018–2019 cohort (0.085 Spearman’s rho, *p*-value = 0.534).

### 3.3. Reconstructive Techniques and Follow-Up

Various reconstructive techniques were employed, with local flap reconstruction being the most frequent, which was used in 71 cases (46.4%). Dermal regeneration templates (DRT) were utilized in 52 cases (34%), and primary intention closure was chosen in 30 cases (19.6%). During the follow-up period, which averaged 9.13 ± 12.33 months, local recurrence was observed in 20 patients (13.1%). The survival analysis for local recurrence revealed that the median survival time was undefined for both periods, as the survival curves did not drop below 50% during the observation. Moreover, the restricted mean survival times were similar, indicating comparable average durations of recurrence-free survival. The Cox regression model showed a hazard ratio of 0.13 (95% CI: 0.01–1.31, *p* = 0.084) for the pandemic period compared to pre-pandemic, suggesting a lower risk of local recurrence, though this result was not statistically significant. The 12-month survival rates were 79% for the pre-pandemic period and 100% for the pandemic period, at 24 months The log-rank test indicates a *p*-value of 0.051, suggesting no statistically significant difference in survival probabilities between the two periods, although the *p*-value is close to significance, as shown in Figure 2.

## 4. Discussion

This retrospective cohort study encompassed 111 patients diagnosed with NMSC of the head and neck region treated by a maxillofacial surgery unit. The patient distribution aligns with the epidemiological profile of NMSC, which typically manifests in older individuals due to cumulative sun exposure over time and is more prevalent in males [2]. According to NMSC pathogenesis, the most frequently affected anatomical site in this cohort was the scalp [15]. The mean tumor size was 2.36 ± 1.7 cm and while the majority of cases were classified as T1 (53.2%) according to the TNM clinical classification (indicative of early-stage lesions), a consistent remnant proportion of cases were T2 and T3, requiring large excision and therefore more invasive management [16]. Lymph node metastasis was found in 2% of patients (three cases), in accordance with data from the literature [17]. Histologically, SCC emerged as the predominant subtype, accounting for 54.4% of cases, followed by BCC at 41.2%. This distribution concurs with the prevalence rates of these histotypes in NMSC cases, even though the higher number of SCC cases over BCC should be interpreted with caution given the small absolute number of cases and the predestined selection of patients, which are mainly referred to our clinic by other specialties (i.e., dermatology) [18]. The most frequently employed reconstruction strategy was local flap reconstruction followed by dermal regeneration templates (DRTs) and primary intention closure. These diverse reconstructive approaches stem from surgical considerations of tumor size, infiltration, location, and patient-specific factors during surgical planning [19,20]. Local recurrence manifested in 20 patients (13.1%): the Kaplan–Meier survival analysis depicted in Figure 2 suggests a favorable trend in local recurrence-free survival for patients during the pandemic compared to the pre-pandemic period. Despite the observed differences, the log-rank test yielded a *p*-value of 0.051, indicating no statistical significance. This marginal *p*-value may imply that while there were differences in protocols due to the COVID-19 pandemic, these did not substantially alter the overall local recurrence rates of NMSC. Albeit the high survival probability might reflect the healthcare system’s adaptation to ensuring continued quality care during the pandemic, given the small number of events, our results should be interpreted cautiously. We emphasize the necessity for vigilant postoperative monitoring and follow-up care to detect and manage potential recurrences [21,22].

The COVID-19 outburst has profoundly altered daily aspects of surgical practice and healthcare in general. The risk of intraoperative transmission, strict adherence to procedural algorithms for patients’ access to the hospital ward and operating theatre, and the fluctuating availability of workforce due to health operator illness had a significant impact on cancer care [23]. This study also brings to light the distinct approaches taken by dermatology and maxillofacial surgery departments during the pandemic. While dermatologists increasingly relied on telemedicine for patient consultations [24], maxillofacial surgeons faced unique challenges in managing in-person treatments with enhanced safety measures. This divergence in strategies highlights the adaptability and resourcefulness of different medical specialties in ensuring continuous patient care amidst unprecedented healthcare disruptions. Moreover, some social aspects need to be taken into account as they can impact pandemics as well as NMSC management. For instance, significant disparities in COVID-19 impacts among different racial and ethnic groups in the United States have been reported [25]. The study reveals that minorities, particularly Hispanic/Latinx, American Indian/Alaskan Native, Native Hawaiian/Pacific Islanders, and Black populations, have been disproportionately affected, with higher incidence and mortality rates during the first phases of the COVID-19 pandemic. Key factors contributing to these disparities include over-representation in essential workforces, increased exposure risks due to housing and environmental conditions, systemic oppression, and limited access to healthcare. This is relevant as we should consider that, according to a recent systematic review [26], the dermatoscopic characteristics of skin cancer across various ethnicities (e.g., Hispanic, Asian, and Black patients) significantly differ from Caucasian characteristics, which is nonetheless the most studied in literature. The review includes 30 records, covering BCC, SCC, and melanoma in various racial groups: given the diversity in dermatoscopic presentations of skin cancers across different races and phototypes, the importance of tailored approaches to diagnosis and management is highlighted.

In response to the pandemic, in March 2020 the Italian health ministry advised the regional administrations to temporarily suspend the deferrable surgical activities, leaving space for emergency/urgent procedures and indispensable surgeries such as for oncologic patients and non-oncologic patients classified as high-risk for worsening of the health conditions [27]. This approach to the COVID-19 pandemic, followed by several other countries, led to a decrease in NMSC diagnosed and treated during the pandemic, compared to the pre-pandemic situation, according to some authors [28,29,30,31]. In a study published in early 2020, the impact of the COVID-19 pandemic on the diagnosis and management of cutaneous malignancies in Australia was investigated using data from the Australian Medicare Benefits Schedule database [28]. The study applied Holt–Winters modeling, a robust forecasting technique, to analyze monthly data spanning from January 2017 to June 2020, stratified into three categories: diagnostic biopsies of skin, excision of NMSC, and excision of melanomas. The models exhibited exceptional fitness and predictive accuracy, as indicated by low mean absolute percentage error values. Significant reductions in clinical and surgical activities were observed during the early months of the pandemic, particularly for NMSC excisions, which declined by 8.1%, 12.2%, and 20.9% in March, April, and May 2020, respectively, when compared to predicted values. In contrast, in a prospective comparative study conducted in a Greek tertiary hospital plastic surgery unit [11], an increase in the number of skin cancer operations in 2020 compared to 2019 was noticed, which was particularly significant for SCC. While there were no significant differences in clinical parameters for NMSC, the type of reconstructive procedures performed changed notably in 2020 with more frequent adoption of skin grafts and flaps procedures over primary intention closure. The authors attributed these findings to the treatment time delay caused by the pandemic, leading to an increase in late-diagnosed NMSC cases and more invasive surgical approaches and worse oncologic outcomes.

Analyzing the results of the present study, an increase in NMSC cases during the pandemic period compared to the pre-pandemic period, was observed. This could be justified by the fact that, despite the potential reluctance of the patient to seek medical attention, the number of dermatologic clinical diagnoses of high-grade head and neck NMSC did not decrease. Consequently, the number of cases referred to the maxillofacial unit during the pandemic was similar to the pre-pandemic period, with a rise in BCC cases statistically significant when using the chi-square test (see Table 2). Another reason for the lack of reduction in diagnosed and treated NMSC cases during the pandemic could be the easier visibility of these lesions, being the head and neck district. Indeed, it is the most exposed area of the body, hence raising concern and seeking early medical consultation despite the COVID-19 pandemic.

However, the effects of COVID-19 outburst were noted in the surgical practice routine, with a statistically significant increase in procedures performed under local anesthesia (LA) exceeding those performed under general anesthesia. There was also a statistically significant reduction in the mean surgical time from 159.333 ± 100.097 min in the pre-pandemic period to 125.415 ± 103.848 min in the pandemic period. Moreover, the surgical time strictly correlated with the dimensions of the excised lesion during the pandemic period, and therefore to the extent of reconstruction. The reduction in mean surgical time could be explained by the adhesion of the surgical theatre operators to the strict indications of national and local guidelines, optimizing the procedures to reduce the contact time with the patient. This shift of approach to the anesthesiologic aspect of surgery during the COVID-19 pandemic underscores the deep collaboration between surgeon and anesthesiologist. They need to be able to change their usual routine to adapt to new challenges such as minimizing contact with the patient while optimizing human resources and maintaining the same quality standards. This is made possible by developing prioritization schedules based on the impact of surgery on survival, function, life expectancy, and available resources [32], but also by investing in the surgeon–anesthesiologist relationship, which has a critical role in the working environment and in patient safety [14,33,34]. In our analysis, we observed a notable shift in anesthesia patterns during the pandemic period. This change can be primarily attributed to alterations in hospital protocols and heightened patient safety measures implemented in response to COVID-19. Studies by Mejía-Terrazas et al. (2021) and Hotta (2021) highlight the advantages of local anesthesia in this context [35,36]. Unlike general anesthesia, which requires airway intervention and may exacerbate COVID-19 pneumonia, LA is not an aerosol-generating procedure, thereby reducing the risk of virus transmission to medical staff. The use of LA, particularly neuraxial anesthesia, has been shown to have minimal adverse effects on clinical outcomes in COVID-19 patients. However, it is crucial to plan surgeries such that they can be completed entirely under LA to avoid the risks associated with unplanned conversion to general anesthesia. In this context, ensuring that the procedure is performed by experienced physicians can minimize the incidence of failed blocks and the use of long-acting local anesthetics can prolong the anesthetic effect, enhancing patient safety and comfort. As reported in a recent systematic review, the pandemic has necessitated the modification of preoperative evaluations and the establishment of comprehensive perioperative care strategies, including the prioritization of LA where feasible [37]. This approach not only benefits patients by minimizing potential complications but also plays a critical role in protecting healthcare personnel from exposure to the virus.

As mentioned, we observed a stronger correlation between surgical time and tumor dimension during the pandemic (see Figure 1), which might initially seem paradoxical given the similar tumor dimensions across both periods. This phenomenon can be attributed to several factors specific to the pandemic context. Firstly, the pandemic brought about a heightened focus on preoperative planning and caution in surgical procedures driven by the need to minimize complications and avoid re-admissions, which could have led to a more meticulous approach to surgery, particularly for larger tumors. Additionally, the aforementioned adaptations in surgical teams and workflows during the pandemic, such as changes in staffing, use of personal protective equipment, enhanced surgeon–anesthesiologist cooperation, and adoption of LA, could have influenced the efficiency and dynamics of surgical procedures. These changes might have made the operation duration more sensitive to tumor dimensions than in the pre-pandemic period.

When comparing the TTI, time from the first visit to surgery, an increment of days without significant changes was found between the two cohorts. In this comparison, a relevant incidence of absent data in the pre-pandemic cohort needs to be stressed. A previous study with a similar design was retrieved from the international literature; this study investigated the impact of the COVID-19 pandemic on the surgical management of NMSC of the head and neck in elderly patients treated in a tertiary referral center in London, United Kingdom [12]. The analysis encompassed 520 cases, including BCC and SCC treated during the pandemic (January 2020 to January 2021), and compared these to cohorts from pre-pandemic period. The study revealed a conspicuous delay in the TTI during the pandemic, averaging 119 days, exceeding national guidelines. Common reconstructive techniques included local flaps and direct skin closure, while DRTs and skin grafts were also utilized. The study underscored the critical importance of timely skin cancer intervention, even amid the pandemic, and highlighted the necessity of minimizing infection risks, particularly for vulnerable populations such as elderly and immunocompromised individuals. The study proposed several strategies, including the implementation of teledermatology, the establishment of “green pathways” for COVID-19-negative patients, and the preferential use of local anesthesia when possible, to enhance patient care while effectively managing the challenges posed by the COVID-19 pandemic [13]. Considering the previous literature together with our findings, we stress the critical importance of timely diagnosis in skin cancer management. The pandemic posed unique challenges in this regard, potentially leading to delayed diagnoses and treatments. It is mandatory to maintain efficient diagnostic processes to prevent worsened outcomes, emphasizing the balance between pandemic-related healthcare constraints and the imperatives of cancer care.

This study has limitations to note and acknowledge when interpreting the results. Firstly, the retrospective design may have introduced inherent selection bias, potentially impacting the robustness of our findings. Secondly, it is a monocentric study with a relatively small sample size—despite being comparable to similar studies in this field [13]—that may limit the generalizability of our findings. Additionally, a potential selection bias arises from the fact that most patients are referred to the maxillofacial surgery unit by dermatology units of private dermatologists. It is conceivable that, due to the pandemic, the approach of dermatologists to NMSC may differ from the pre-COVID-19 period, possibly resulting in a reduction in the presentation of non-cancerous illnesses and an increased tendency for dermatologists to autonomously treat more head and neck skin cancer cases. Despite this, we believe that our findings are still largely reflective of the overall impact of the COVID-19 pandemic on the diagnosis and management of NMSC in the head and neck region. Moreover, this study contributes to the wider field of oncological surgery, serving in this context for exploring future challenges and innovations as the healthcare system continues to adapt post-pandemic. The COVID-19 has underscored the critical need for resilient healthcare strategies, highlighting the necessity for flexible, robust approaches in oncological surgery to ensure sustained high-quality care amidst future healthcare challenges [38]. Our results will inform strategies to maintain the level of care in the face of potential future healthcare disruptions [39]. 

## 5. Conclusions

This study did not show a reduction in the number of treated NMSC cases during the pandemic period, possibly due to several factors, including the visibility of new lesions when growing in the head and neck district. However, the impact of COVID-19 deeply changed the daily practice of surgeons, posing new challenges to offer the best healthcare while preventing the further spread of the viral disease. Indeed, our study demonstrated a noteworthy reduction in the mean surgical time during the pandemic period and a significant increase in cases treated under local anesthesia rather than general anesthesia. These findings express the importance of collaboration among all healthcare workers inside the surgical theatre who need to conform to the national guidelines while optimizing their resources and aiming to provide the best care for the patient.

## Figures and Tables

**Figure 1 healthcare-12-00501-f001:**
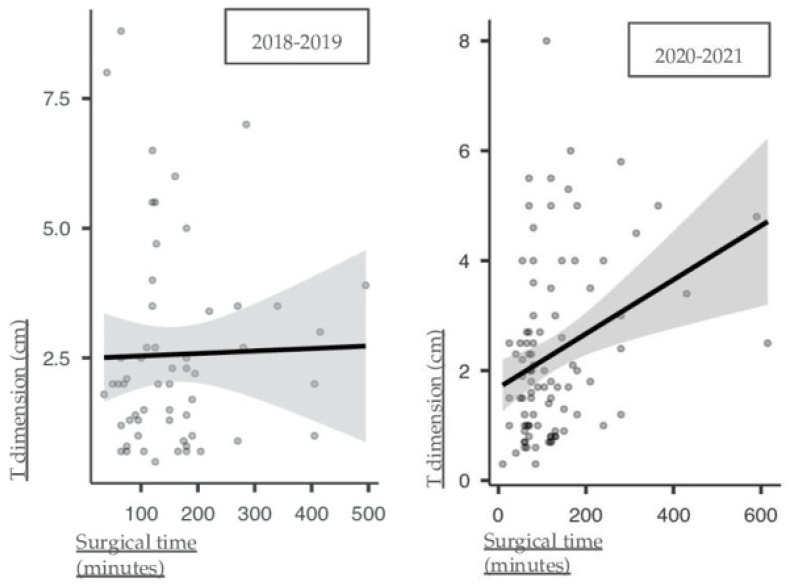
Correlation matrix of surgical time and cancer dimension in the 2018–2019 (0.085 Spearman’s rho, *p*-value = 0.534) and 2020–2021 (0.315 Spearman’s rho, *p*-value = 0.002) cohorts.

**Figure 2 healthcare-12-00501-f002:**
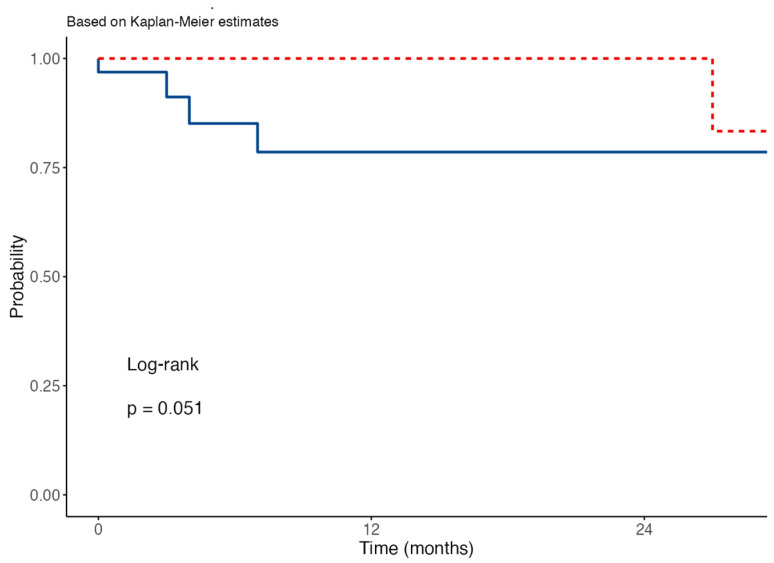
Kaplan–Meier survival curves for local recurrence of non-melanoma skin cancer across two periods. Legend: The Kaplan–Meier survival curves depict the probability of survival free from local recurrence of non-melanoma skin cancer over a 24-month period for two cohorts. The pre-pandemic period (blue solid line) represents patients before the COVID-19 pandemic, and the pandemic period (red dashed line) represents patients during the pandemic. The log-rank test indicates a *p*-value of 0.051, suggesting no statistically significant difference in the survival probabilities between the two periods.

**Table 1 healthcare-12-00501-t001:** Frequencies of location in different subunits.

Head Subunits	Counts (% of Total)	Cohort	Counts (% of Total)
Frontal	23 (14.4%)	2018–2019	9 (5.6%)
		2020–2021	14 (8.8%)
Nose	28 (17.5%)	2018–2019	7 (4.4%)
		2020–2021	21 (13.1%)
Palpebral	5 (3.1%)	2018–2019	4 (2.5%)
		2020–2021	1 (0.6%)
Cheek	29 (18.1%)	2018–2019	9 (5.6%)
		2020–2021	20 (12.5%)
Superior lip	3 (1.9%)	2018–2019	2 (1.3%)
		2020–2021	1 (0.6%)
Inferior lip	9 (5.6%)	2018–2019	7 (4.4%)
		2020–2021	2 (1.3%)
Mental	1 (0.6%)	2018–2019	1 (0.6%)
		2020–2021	0 (0.0%)
Ear	15 (9.4%)	2018–2019	5 (3.1%)
		2020–2021	10 (6.3%)
Scalp	46 (28.7%)	2018–2019	19 (11.9%)
		2020–2021	27 (16.9%)
Neck	1 (0.6%)	2018–2019	0 (0.0%)
		2020–2021	1 (0.6%)

**Table 2 healthcare-12-00501-t002:** Descriptives and chi-square test of categorical data.

	2018–2019 (63 Cases) Counts (% of Total)	2020–2021 (97 Cases) Counts (% of Total)	*p*-Value
Female	17 (10.6%)	35 (21.9%)	0.304
Male	46 (28.7%)	62 (38.8%)	
Local Anesthesia	18 (11.3%)	55 (36.5%)	0.001
General Anesthesia	39 (25.8%)	40 (26.5%)	
BCC	20 (12.5%)	46 (28.7%)	0.037
SCC	42 (26.3%)	45 (28.1%)	
Negative Margin	51 (32.9%)	69 (44.5%)	0.197
Positive Margin	10 (6.5%)	25 (16.1%)	
Recurrence no	50 (32.7%)	83 (54.2%)	0.379
Recurrence yes	12 (7.8%)	8 (5.2%)	

Abbreviations: BCC (Basal Cell Carcinoma); SCC (Squamous Cell Carcinoma).

**Table 3 healthcare-12-00501-t003:** Descriptives and Mann–Whitney U-test of continuous data.

	Cohort	N	Missing	Mean	Median	SD	Range	Shapiro–Wilk		*p* Value
								w	*p*	
Age	2018–2019	59	4	77.898	82	11.427	50–93	0.916	< 0.001	0.244
	2020–2021	97	0	80.381	82	11.476	45–103	0.953	0.002	
Dimension (cm)	2018–2019	62	1	2.489	2.000	1.881	0.5–8.8	0.830	< 0.001	0.571
	2020–2021	95	2	2.273	1.800	1.583	0.3–8	0.888	<0.001	
TTI (days)	2018–2019	17	46	34.235	28.000	19.412	3–78	0.942	0.338	0.398
	2020–2021	76	21	41.461	35.000	28.414	2–163	0.856	< 0.001	
Surgery time (min)	2018–2019	57	6	159.333	127	100.097	35–495	0.854	<0.001	0.003
	2020–2021	94	3	125.415	87.500	103.848	10–615	0.722	< 0.001	
Margin distance (cm)	2018–2019	52	11	0.376	0.300	0.375	0.000–2	0.777	<0.001	0.465
	2020–2021	94	3	0.318	0.200	0.282	0.000–1.1	0.844	< 0.001	
DOI	2018–2019	37	26	5.292	4.500	3.968	0.000–16	0.912	0.007	
	2020–2021	81	16	4.278	3.000	3.998	0.000–23	0.811	< 0.001	0.098

Abbreviations: DOI (Depth of Invasion); TTI (Time to Treatment Interval).

## Data Availability

Data are contained within the article.

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
