# Peer review of "Impact of the COVID-19 Pandemic on the Diagnosis and Management of Non-Melanoma Skin Cancer in the Head and Neck Region: A Retrospective Cohort Study"

_healthcare, 2024, doi:10.3390/healthcare12040501_

Round 1

Reviewer 1 Report

Comments and Suggestions for Authors

There is no new information in the article.

The method is incomplete.

The sample size is low, so it is not enough for conclusions and generalizability. 

The manuscript contains fundamental errors that cannot be rectified through author revisions.

Author Response

Manuscript ID: healthcare-2783974 - Title: Impact of the COVID-19 Pandemic on Diagnosis and Management of Non-Melanoma Skin Cancer in the Head and Neck Region: A Retrospective Cohort Study

Dear Editor-in-Chief, thanks for reconsidering our manuscript, please find our revised version enclosed. 

We have also attached a list of all the changes we have made, with a reply to the Reviewers.

Sincerely,

The authors of the manuscript.

Correspondence: 

[email protected]

Changes made in the original manuscript in response to the reviewers have been highlighted in yellow. 

Response to Reviewer 1

There is no new information in the article.

We acknowledge your concern regarding the novelty of the information presented. However, we emphasize that our study offers preliminary insights in an area with limited existing literature (e.g. to the best of our knowledge there are no reviews on this theme, just few case reports and retrospective study). Therefore, our findings contribute to the evolving understanding of healthcare dynamics during the pandemic, particularly in the context of non-melanoma skin cancer of the head and neck.

The method is incomplete.

Our methodology was carefully designed and reported in accordance with the STROBE (Strengthening the Reporting of Observational Studies in Epidemiology) guidelines, to ensure rigor and transparency. We believe our methods are adequately detailed, but we are open to expanding on any specific aspects you find lacking.

The sample size is low, so it is not enough for conclusions and generalizability. 

The concern regarding our sample size is acknowledged. We understand that a larger sample size could potentially reduce error margins and enhance the generalizability of our findings in the limitation section of discussion. Specifically, a larger sample size would help minimize Type II errors (false negatives), where true effects might not be detected: we considered this in our conclusions. Our sample size is comparable to similar studies in this domain, like the one by Seretis et al. We have emphasized this comparison in the revised manuscript and have discussed how future studies could build on our findings with a broader sample.

The manuscript contains fundamental errors that cannot be rectified through author revisions.

You mentioned the manuscript contains fundamental errors that cannot be rectified. We respectfully request specific details regarding these errors, as your insights would be invaluable for us to understand and address these concerns. Without specific examples, it is challenging for us to make the necessary corrections.

Reviewer 2 Report

Comments and Suggestions for Authors

I read with great interest the study on impact of Covid-19 era in NMSC approach :

A few concerns:

1) differences between a dermatology and maxilla surgery departments shoukd be outlined and descripted more excessively on how different is the appraoch between the appproach of those departments

2) a reference in NMSC patients in skin of colour and social disparities that intensified in Covid-19 era 

Raine S, Liu A, Mintz J, Wahood W, Huntley K, Haffizulla F. Racial and Ethnic Disparities in COVID-19 Outcomes: Social Determination of Health. Int J Environ Res Public Health. 2020;17(21):8115. Published 2020 Nov 3. doi:10.3390/ijerph17218115

Karampinis E, Lallas A, Lazaridou E, Errichetti E, Apalla Z. Race-Specific and Skin of Color Dermatoscopic Characteristics of Skin Cancer: A Literature Review. Dermatol Pract Concept. 2023 Oct 1;13(4 S1). doi: 10.5826/dpc.1304S1a311S. PMID: 37874992.

3) the importance of timely diagnosed skin cancer should be more outlined

4) nice discussion section 

Author Response

Manuscript ID: healthcare-2783974 - Title: Impact of the COVID-19 Pandemic on Diagnosis and Management of Non-Melanoma Skin Cancer in the Head and Neck Region: A Retrospective Cohort Study

Dear Editor-in-Chief, thanks for reconsidering our manuscript, please find our revised version enclosed. 

We have also attached a list of all the changes we have made, with a reply to the Reviewers.

Sincerely,

The authors of the manuscript.

Correspondence: 

[email protected]

Changes made in the original manuscript in response to the reviewers have been highlighted in yellow.

Response to Reviewer 2

I read with great interest the study on impact of Covid-19 era in NMSC approach :

A few concerns:

1) differences between a dermatology and maxilla surgery departments shoukd be outlined and descripted more excessively on how different is the appraoch between the appproach of those departments

Dear reviewer, thanks for considering our work. In response to your first concern, we have expanded the discussion sections of our manuscript to more thoroughly describe and compare the approaches of dermatology and maxillofacial surgery departments. This includes detailing how each department's methodologies and patient management strategies differed during the COVID-19 era, thus providing a clearer understanding of their unique challenges and responses.

2) a reference in NMSC patients in skin of colour and social disparities that intensified in Covid-19 era Raine S, Liu A, Mintz J, Wahood W, Huntley K, Haffizulla F. Racial and Ethnic Disparities in COVID-19 Outcomes: Social Determination of Health. Int J Environ Res Public Health. 2020;17(21):8115. Published 2020 Nov 3. doi:10.3390/ijerph17218115; Karampinis E, Lallas A, Lazaridou E, Errichetti E, Apalla Z. Race-Specific and Skin of Color Dermatoscopic Characteristics of Skin Cancer: A Literature Review. Dermatol Pract Concept. 2023 Oct 1;13(4 S1). doi: 10.5826/dpc.1304S1a311S. PMID: 37874992.

Acknowledging the critical importance of this aspect, we have incorporated the suggested references into our manuscript. These additions enrich the discussion and provides a more comprehensive view of the social determinants of health that were accentuated during the pandemic.

3) the importance of timely diagnosed skin cancer should be more outlined

We have revised our manuscript to more emphatically outline the importance of timely skin cancer diagnosis. This revision includes a discussion on the challenges and potential consequences of delayed diagnoses during the COVID-19 pandemic, emphasizing the need for prompt and effective management of skin cancer.

4) nice discussion section 

We are pleased to hear that you found the discussion section well-written, thank you for the interest shown in our manuscript.

Reviewer 3 Report

Comments and Suggestions for Authors

Dear editor/author

In this retrospective cohort study, the impact of the COVID-19 Pandemic on the diagnosis and management of non-melanoma skin cancer in the head and neck region was investigated.

The study is generally well-designed. I have a few criticisms about the article. First of all, the results are repeated in the first paragraph in the discussion section. I recommend shortening repetitive information.

It should be discussed in more detail with other studies in the literature.

"Analyzing the results of the present study, an increase in NMSC cases during the pandemic period compared to the pre-pandemic period, even though not statistically significant, was observed." in the discussion section: but in the results section, there was a statistical difference in the number of BCC and SCC during the pandemic period. It was shown that there was a significant increase (p = 0.037). These two statements contradict each other.

Author Response

Manuscript ID: healthcare-2783974 - Title: Impact of the COVID-19 Pandemic on Diagnosis and Management of Non-Melanoma Skin Cancer in the Head and Neck Region: A Retrospective Cohort Study

Dear Editor-in-Chief, thanks for reconsidering our manuscript, please find our revised version enclosed. 

We have also attached a list of all the changes we have made, with a reply to the Reviewers.

Sincerely,

The authors of the manuscript.

Correspondence: 

[email protected]

Changes made in the original manuscript in response to the reviewers have been highlighted in yellow.

Response to Reviewer 3

In this retrospective cohort study, the impact of the COVID-19 Pandemic on the diagnosis and management of non-melanoma skin cancer in the head and neck region was investigated. The study is generally well-designed. I have a few criticisms about the article. First of all, the results are repeated in the first paragraph in the discussion section. I recommend shortening repetitive information.

Dear reviewer, we acknowledge your observation regarding the repetition of results in the first paragraph of the discussion section. To improve the clarity and conciseness of our manuscript, we have revised this section and eliminated most of the redundant information. 

It should be discussed in more detail with other studies in the literature.

Upon your recommendation, we have expanded our discussion to include a more detailed comparison with other studies in the literature. This enhancement provides a broader context to our findings, illustrating how our study resounds from existing research in the field. The following relevant references have been added: Bridgeman SG, Perche PO, Feldman SR. Treatment Adherence in Dermatology During the COVID-19 Pandemic: A Review. Cureus. 2023;15(1):e34141. Published 2023 Jan 24. doi:10.7759/cureus.34141; Raine S, Liu A, Mintz J, Wahood W, Huntley K, Haffizulla F. Racial and Ethnic Disparities in COVID-19 Outcomes: Social Determination of Health. Int J Environ Res Public Health. 2020;17(21):8115. Published 2020 Nov 3. doi:10.3390/ijerph17218115; Karampinis E, Lallas A, Lazaridou E, Errichetti E, Apalla Z. Race-Specific and Skin of Color Dermatoscopic Characteristics of Skin Cancer: A Literature Review. Dermatol Pract Concept. 2023 Oct 1;13(4 S1). doi: 10.5826/dpc.1304S1a311S. PMID: 37874992.; Newlands C, Currie R, Memon A, Whitaker S, Woolford T. Non-melanoma skin cancer: United Kingdom National Multidisciplinary Guidelines. J Laryngol Otol. 2016;130(S2):S125-S132. doi:10.1017/S0022215116000554; Khalid A, van Essen P, Crittenden TA, Dean NR. The anatomical distribution of non-melanoma skin cancer: a retrospective cohort study of 22 303 Australian cases. ANZ J Surg. 2021;91(12):2750-2756. doi:10.1111/ans.17030.

"Analyzing the results of the present study, an increase in NMSC cases during the pandemic period compared to the pre-pandemic period, even though not statistically significant, was observed." in the discussion section: but in the results section, there was a statistical difference in the number of BCC and SCC during the pandemic period. It was shown that there was a significant increase (p = 0.037). These two statements contradict each other.

Dear reviewer, thanks for pointing out the apparent contradiction between the discussion and results sections. This has been corrected in the discussion section to accurately reflect the data presented in the results section, specifically the raise in BCC cases. 

Reviewer 4 Report

Comments and Suggestions for Authors

I was invited to revise the paper entitled "Impact of the COVID-19 Pandemic on Diagnosis and Management of Non-Melanoma Skin Cancer in the Head and Neck Region: A Retrospective Cohort Study". It was a retrospective cohort study from an Italian hospital aimed to evaluate differences in NMSC of the head between pre pandemic and pandemic period. 

The topic is interesting and poor paper on this topic were published on my knowledge.

Observations:

- In introduction section Authors should report the impact of Covid-19 pandemic on diseases managements. Several papers, also from Italy, reported the changed in admissions rate for oncological and non-oncological diseases during pandemic;

- About methods, Authors should tests continous variables for normality distribution. For example, as reported in table 3, Age seems to be normally distributedm so it should be analyzed with parametric tests;

- Table 1 should be reported dividing cancer site also by study periods;

- Table 2 should report also percentage of each variables;

- Under Table 3, abbreviations used should be reported in order to improve readability;

- How did Authors explaine the different pattern of anesthesia during pandemic? 

- Why surgery times were higher during pandemic period despite the dimension of tumors were similar? did cancer appeared worst due to a delay in diagnosis/surgery? 

- How did Authors explain the strongest correlation between time and dimension during pandemic despite a lack in differences among dimensions?

- I suggest to perform a survival analysis on reccurrence to evaluate possible difference in recurrence time between study groups;.

Author Response

Reviewer 4

I was invited to revise the paper entitled "Impact of the COVID-19 Pandemic on Diagnosis and Management of Non-Melanoma Skin Cancer in the Head and Neck Region: A Retrospective Cohort Study". It was a retrospective cohort study from an Italian hospital aimed to evaluate differences in NMSC of the head between pre pandemic and pandemic period. 

The topic is interesting and poor paper on this topic were published on my knowledge.

Dear reviewer, thanks for reading and appreciating our work.

Observations:

- In introduction section Authors should report the impact of Covid-19 pandemic on diseases managements. Several papers, also from Italy, reported the changed in admissions rate for oncological and non-oncological diseases during pandemic; 

The introduction does mention the impact of COVID-19 on healthcare systems and the postponement or modification of elective medical procedures, including those for NMSC. It also references previous studies on the impact of the pandemic on treatment delay for NMSC​​. Nonetheless, we added a paragraph with more data retrived in the literature.

- About methods, Authors should tests continous variables for normality distribution. For example, as reported in table 3, Age seems to be normally distributedm so it should be analyzed with parametric tests;

We have conducted normality tests for continuous variables and included the results in Table 3. The Shapiro-Wilk test indicated non-normal distributions for most variables, and we used appropriate statistical methods (non-parametric tests) for the analysis.

- Table 1 should be reported dividing cancer site also by study periods;

We have revised Table 1 to include a division of cancer site data by pre-pandemic and pandemic study periods, providing a clearer comparison.

- Table 2 should report also percentage of each variables;

We have updated Table 2 to include the percentage of each variable, enhancing the comprehensiveness of our data presentation.

- Under Table 3, abbreviations used should be reported in order to improve readability;

We have added a list of abbreviations under Table 3 to improve readability and ensure clarity for all readers.

- How did Authors explaine the different pattern of anesthesia during pandemic? 

We have included a discussion on the observed shift in anesthesia patterns during the pandemic, considering factors such as hospital protocols and patient safety measures.

- Why surgery times were higher during pandemic period despite the dimension of tumors were similar? did cancer appeared worst due to a delay in diagnosis/surgery? 

In response to your question about the surgery times during the pandemic period, our data interestingly revealed that surgery times were, in fact, significantly shorter during the pandemic, despite similar tumor dimensions compared to the pre-pandemic period. This finding could be attributed to several factors that influenced surgical practice during the COVID-19 pandemic, as we deeply discussed within the manuscript.

- How did Authors explain the strongest correlation between time and dimension during pandemic despite a lack in differences among dimensions?

The stronger correlation between surgical time and tumor dimension during the pandemic likely reflects a combination of increased surgical precision and the unique operational challenges and adaptations in the surgical environment during the COVID-19 pandemic; we reported this view in the discussion section.

- I suggest to perform a survival analysis on reccurrence to evaluate possible difference in recurrence time between study groups. 

Dear reviewer, we condnucted the survival analysis on reccurrence as requested and no significant differences were retrieved as now reported in the study.

Round 2

Reviewer 1 Report

Comments and Suggestions for Authors

-

Comments on the Quality of English Language

-

Author Response

-

Reviewer 3 Report

Comments and Suggestions for Authors

Dear Editor,

I think the revised version of the article can be accepted.

Author Response

Dear reviewer, thanks for appreciating our revised version of the manuscript.

Reviewer 4 Report

Comments and Suggestions for Authors

Authors addressed properly the great part of my previous observations. In my opinion minor revisions should be implemented.

- in introduction section, Authors should report the impact of pandemic on oncological surgery in general and after they should focus on NMSC;

- both in introduction section and discussion section, Authornshould emphatize the importance of this study;

- among discussions, comparing these results with also other kind of cancers, Authors should also discuss next issues in oncological surgery after pandemic.

Author Response

Dear reviewer, thanks for reeding our revised version of the manuscript. 

  • in introduction section, Authors should report the impact of pandemic on oncological surgery in general and after they should focus on NMSC.
  • According to your suggestion we expanded in the introduction section the impact of pandemic in oncological surgery. We added a reference and the findings of a recent large population study conducted in USA (lines 52-58).
  • both in introduction section and discussion section, Author should emphatize the importance of this study;
  • We have emphasized the importance of our study both in the introduction (lines 71-75) and in the discussion section (lines 347-353).
  • among discussions, comparing these results with also other kind of cancers, Authors should also discuss next issues in oncological surgery after pandemic.
  • We have further discussed accordingly our findings (lines 347-353)